# Exposure to the Amino Acids Histidine, Lysine, and Threonine Reduces mTOR Activity and Affects Neurodevelopment in a Human Cerebral Organoid Model

**DOI:** 10.3390/nu14102175

**Published:** 2022-05-23

**Authors:** Amber Berdenis van Berlekom, Raphael Kübler, Jeske W. Hoogeboom, Daniëlle Vonk, Jacqueline A. Sluijs, R. Jeroen Pasterkamp, Jinte Middeldorp, Aletta D. Kraneveld, Johan Garssen, René S. Kahn, Elly M. Hol, Lot D. de Witte, Marco P. Boks

**Affiliations:** 1Department of Psychiatry, University Medical Center Utrecht, Brain Center, Utrecht University, 3584 CG Utrecht, The Netherlands; a.berdenisvanberlekom-2@umcutrecht.nl (A.B.v.B.); rene.kahn@mssm.edu (R.S.K.); lotje.dewitte@mssm.edu (L.D.d.W.); 2Department of Translational Neuroscience, University Medical Center Utrecht, Brain Center, Utrecht University, 3584 CG Utrecht, The Netherlands; jeske_hoogeboom@hotmail.com (J.W.H.); d.n.vonk-3@umcutrecht.nl (D.V.); j.a.sluijs@umcutrecht.nl (J.A.S.); r.j.pasterkamp@umcutrecht.nl (R.J.P.); middeldorp@bprc.nl (J.M.); e.m.hol-2@umcutrecht.nl (E.M.H.); 3Department of Psychiatry, Icahn School of Medicine at Mount Sinai, New York, NY 10029, USA; raphael.kubler@mssm.edu; 4Department Neurobiology & Aging, Biomedical Primate Research Centre, 2288 GJ Rijswijk, The Netherlands; 5Division of Pharmacology, Utrecht Institute for Pharmaceutical Sciences, Faculty of Science, Utrecht University, 3584 CG Utrecht, The Netherlands; a.d.kraneveld@uu.nl (A.D.K.); johan.garssen@danone.com (J.G.); 6Danone Nutricia Research, 3584 CT Utrecht, The Netherlands; 7Mental Illness Research, Education, and Clinical Center (VISN 2 South), James J Peters VA Medical Center, Bronx, NY 10468, USA

**Keywords:** neurodevelopment, mTOR, amino acids

## Abstract

Evidence of the impact of nutrition on human brain development is compelling. Previous in vitro and in vivo results show that three specific amino acids, histidine, lysine, and threonine, synergistically inhibit mTOR activity and behavior. Therefore, the prenatal availability of these amino acids could be important for human neurodevelopment. However, methods to study the underlying mechanisms in a human model of neurodevelopment are limited. Here, we pioneer the use of human cerebral organoids to investigate the impact of amino acid supplementation on neurodevelopment. In this study, cerebral organoids were exposed to 10 mM and 50 mM of the amino acids threonine, histidine, and lysine. The impact was determined by measuring mTOR activity using Western blots, general cerebral organoid size, and gene expression by RNA sequencing. Exposure to threonine, histidine, and lysine led to decreased mTOR activity and markedly reduced organoid size, supporting findings in rodent studies. RNA sequencing identified comprehensive changes in gene expression, with enrichment in genes related to specific biological processes (among which are mTOR signaling and immune function) and to specific cell types, including proliferative precursor cells, microglia, and astrocytes. Altogether, cerebral organoids are responsive to nutritional exposure by increasing specific amino acid concentrations and reflect findings from previous rodent studies. Threonine, histidine, and lysine exposure impacts the early development of human cerebral organoids, illustrated by the inhibition of mTOR activity, reduced size, and altered gene expression.

## 1. Introduction

Compelling evidence from epidemiological studies suggests that the maternal diet during pregnancy is a key modifier of neurodevelopment and impacts later-life intelligence, social function, and the risk of acquiring a range of neuropsychiatric conditions, such as autism spectrum disorders and schizophrenia [1,2,3,4,5,6]. Particularly during early gestation, brain development is vulnerable to maternal nutritional deviations, with effects persisting later in life. Pre-clinical studies on prenatal nutrition, the availability of micronutrients, and the composition of the maternal diet have shown a broad range of effects in offspring, such as decreased neurogenesis, changes in neuronal dendritic arborization, and increased astrocytic GFAP expression [7,8,9,10].

Amino acids are a key component of nutrition, and some essential amino acids need to be provided through the diet. Dietary intake influences plasma amino acid concentrations and ratios [11], and supplementation with specific amino acids is used to improve health, metabolism, and athletic performance [12]. Amino acids are best known as the building blocks of proteins but also have an important regulatory function in the cell [12].

Recently, amino acids have emerged as potent modulators of the mammalian target of rapamycin (mTOR) [13,14,15]. mTOR signaling regulates the phosphorylation of the translational modulator P70S6K and is involved in processes such as cell growth, metabolism, and autophagy but also in neurodevelopment, the regulation of cortical structure formation through outer radial glia, the timing of the gliogenic switch, and axon formation and dendritic arborization [16,17,18,19,20]. Deregulation of mTOR function due to genetic mutations or altered protein expression is involved in brain diseases, particularly developmental neuropsychiatric disorders such as autism, schizophrenia, and tuberous sclerosis [18,21,22,23,24].

Previous in vitro studies showed that supplementation with three specific amino acids, histidine (His), lysine (Lys), and threonine (Thr), synergistically potently inhibit P70S6K signaling downstream of mTROC1 and thereby protein synthesis and modulate IRS-1 phosphorylation [25,26]. Furthermore, a mouse study enriching the postnatal diet of 5-week-old mice with these three essential amino acids found decreased mTOR activity in the mouse brains and an effect on autism-related behaviors [26].

Therefore, amino acid availability, especially that of His, Lys, and Thr, at early developmental stages could be important for healthy brain development, and dietary changes during pregnancy may have consequences for the risk of neurodevelopmental disorders.

To date, studies on the role of amino acids in mTOR signaling have been performed in mouse models and 2D in vitro studies in rodent mammary epithelial cells [25,26]. Because of the vast differences between human and rodent neurodevelopment and cell function [27,28,29], translation to a human model is important, as it may be more accurate in reporting transcriptional, functional, and neurodevelopmental changes. The introduction of the induced pluripotent stem cell (iPSC)-derived cerebral organoid model by Lancaster et al. (2013) has provided an opportunity to study human neurodevelopment in vitro [30,31]. Cerebral organoids show structural properties that are specific to the developing human cortex, such as the presence of an outer subventricular zone containing outer radial glia [31,32]. Furthermore, as cerebral organoids mature, they contain a multitude of different cell types, which form an integrated network, as opposed to single-cell-type 2D in vitro approaches [31,33,34]. Interestingly, with slight modifications to the cerebral organoid protocol, the presence of microglia was established [33]. This is unique compared to other brain organoid models and allows for the study of immune-related phenotypes [33]. Altogether, cerebral organoids provide a useful representation of early human cortical neurodevelopment at transcriptomic, epigenetic, and structural levels [31,33,35,36]. In addition, this in vitro model is a more versatile and accessible resource for sample material than human primary early gestational brain tissue.

This study pioneers in researching the effects of His, Lys, and Thr exposure on mTOR activity in a human neurodevelopmental 3D in vitro model. The primary need for the study is two-fold: 1. Establish whether we can replicate findings on mTOR activity upon His, Lys, and Thr exposure from previous in vitro and in vivo work using the human cerebral organoid model. 2. Explore neurodevelopmental responses to the inhibitory effects of His, Lys, and Thr on mTOR activity in human cerebral organoids by investigating changes in general size and gene expression. The developing cerebral organoids were exposed to increased concentrations of the three amino acids His, Lys, and Thr (AA exposure) and assessed for mTOR activity by analyzing Western blots, general organoid size, and transcriptomic alterations using RNA sequencing. The data were cross-referenced with gene lists on biological processes and cell types to determine leads for future research.

## 2. Materials and Methods

### 2.1. iPSC line Generation and Maintenance

The generation and characterization of induced pluripotent stem cell (iPSC) lines OH1.5 (male, 62 years old), OH2.6 (male, 61 years old), and OH4.6 (female, 60 years old) were performed in the MIND facility of the UMC Utrecht, described previously [33,37]. iPSC lines were maintained feeder-free on Geltrex (Thermofisher, A1413202, Thermo Fisher Scientific, Waltham, MA, USA) in StemFlex8 medium (ThermoFisher, A3349401) at 37 °C with 5% CO_2_. Medium was changed 3 times a week. Cells were passaged once a week at 80–90% confluency by incubating the cells in 5 μM EDTA (ThermoFisher, 15575020) for 2 min at 37 °C and transferring cell aggregates to a new culture dish with StemFlex8 medium, supplemented with 5 μM ROCK inhibitor (Axon, 1683, Axon Medchem BV, Groningen, The Netherlands) for the first 24 h. All lines were kept in culture for a maximum of 60 passages and were regularly tested for mycoplasma infections (Lonza, LT07-318, Lonza Bioscience Solutions, Basel, Switzerland).

### 2.2. Ethical Approval

For iPSC line generation, written informed consent was provided by volunteers (without neurodevelopmental, psychiatric, neurologic, or genetic disorders), and approval was granted by the Medical Ethical Committee of the University Medical Center Utrecht. This study was conducted in accordance with the Code of Ethics of the World Medical Association (Declaration of Helsinki).

### 2.3. Organoid Differentiation

Human iPSCs were differentiated towards cerebral organoids as described previously [33]. In short, iPSCs were grown until ~90% confluency and dissociated to single cells using 5 μM EDTA followed by Accutase (Thermofisher, A11105-01). Cells were counted with the Countess™ II FL Automated Cell Counter (ThermoFisher) and plated at a concentration of 3.5 × 10^6^ cells per well in an aggrewell800 microwell plate (StemCell Technologies, 27865) in 2 mL of hES0 medium (DMEM-F12 (ThermoFisher, 11320-074), 20% KOSR (ThermoFisher, 10828028), 1% NEAA (ThermoFisher, 11140-035), 1% L-Glutamine (ThermoFisher, 25030-024), 3% FBS (SigmaAldrich, F7524, Sigma-Aldrich, Saint Louis, MO, USA), 496 μM ß-mercaptoethanol (Merck-Schuchardt, 805740, Merck Schuchardt oHG, Hohenbrunn, Germany) supplemented with 4 ng/mL bFGF (ThermoFisher, AA10-155) and 50 μM ROCK inhibitor (Axon Medchem, Y-27632). Medium was refreshed on day 1. On day 2, embryoid bodies were transferred to an ultra-low attachment 96-well plate (Corning, 3474, Corning, Corning Inc., NY, USA). Medium was replaced with hES0 without ROCK inhibitor and bFGF on day 4. On day 6, medium was replaced with neural induction medium (DMEM-F12, 1% N2 (ThermoFisher, 17502048), 1% L-Glutamine, 1% NEAA, and 0.1 µg/mL heparin (Sigma Aldrich, H3149-10KU)), which was refreshed on days 8, 10, and 12. Organoids were embedded in 30 μL of Matrigel (Corning, 356234) on day 13 and cultured in cerebral organoid differentiation medium without vitamin A (DMEM-F12 1:1 with neurobasal medium (ThermoFisher, 21103049), 1% L-Glutamine, 1% P/S, 0.025% insulin (Sigma Aldrich, I9278), 3.5 μL/L 2-mercaptoethanol, 1% NEAA, and 1:100 B27 supplement without vitamin A (Sigma Aldrich, 12587010)) for 4 days, refreshing the medium on day 15 with 16 organoids per 60 mm dish. Medium was replaced with cerebral organoid differentiation medium with vitamin A (DMEM-F12 1:1 with neurobasal medium, 1% L-Glutamine, 1% P/S, 0.025% insulin, 3.5 μL/L 2-mercaptoethanol, 1% NEAA, and 1:100 B27 supplement with vitamin A (Sigma Aldrich, 17504044)) on day 17. The organoids were kept on a belly dancer (BDRAA115S, IBI Scientific, Dubuque, IA, USA) at speed 3 and maintained in cerebral organoid differentiation medium with vitamin A for the duration of the experiment (Figure 1).

### 2.4. Exposure

Before exposure, cerebral organoids grown in separate culture dishes were combined and randomly redivided over separate 60 mm culture dishes for different conditions to reduce intra-dish variation [38]. Cerebral organoids were exposed to 100 nM rapamycin in DMSO (5312-88-9, LC Laboratories, Woburn, MA, USA) or medium with an additional 10 mM or 50 mM of the amino acids Thr (Sigma Aldrich, 1084110010), His (Sigma Aldrich, 1043500025), and Lys (Sigma Aldrich, 1057000100) in a 1:1:1 ratio (AA exposure) in differentiation medium with vitamin A. After supplementation with amino acids, the pH of the medium was adjusted to pH 7.4 with NaOH. Rapamycin exposure was performed at week 4 for 1 h. Short-term AA exposure was performed for 1 h at the start of week 4 to test the acute response to the 3 amino acids. AA exposure continued to week 15, changing the medium 3 times per week with fresh differentiation medium with amino acids. A schematic representation of the timeline can be found in Figure 1. All experiments included a control condition that received standard differentiation medium with vitamin A. To observe the possible inhibitory effect of rapamycin or AA exposure, cerebral organoids were always harvested 1 h after medium change (with either control medium or medium supplemented with rapamycin or the 3 amino acids) for protein and RNA isolation.

### 2.5. Size Measurements

Bright-field pictures (2.5×) were obtained (EVOS M5000 microscope, Thermo Fisher) from organoids that were subjected to chronic AA exposure from the start of week 4 (day 21). Pictures were taken at the end of week 4, week 5, week 6, week 7, and week 10. Organoid size was determined by creating a mask of the images and measuring the area using a macro in FIJI [39]. The masks were all verified manually.

### 2.6. Western Blot

For protein isolation, cerebral organoids were collected individually in 100 µL of suspension buffer (0.1 M NaCl, 0.01 M Tris-HCL pH 7.6, 0.001 M EDTA, complete EDTA-free protease inhibitor cocktail (Roche, 11697498001), and phosphatase inhibitor cocktail (Sigma Aldrich, P5726) and lysed using the Ultra Turrax Homogenizer (0003737000, IKA Werke, Staufen im Breisgau, Germany). Then, 2× SDS loading buffer (100 mM Tris p H6.8, 4% SDS, 20% glycerol, and 0.2 M DTT (Cleland’s reagent, 10708984001)) was added 1:1, and samples were heated at 95 °C for 5 min. DNA was sheared with a 25-gauge needle. Bromophenol Blue (34725-61-6, MERCK, Darmstadt, Germany) was added to each sample for visibility. Samples were stored at −80 °C until further use. Proteins were separated by SDS-PAGE gel electrophoresis on 7.5%, 10%, or 15% gels and blotted on a 0.45 μM pore size nitrocellulose membrane (A20485269, GE Healthcare, Chicago, IL, USA) using wet blotting. Blots were incubated in blocking buffer (50 mM Tris pH 7.4, 150 mM NaCl, 0.25 (*w*/*v*) gelatin, and 0.5% triton X-100)) and incubated with fluorescent primary antibody (Appendix A) in blocking buffer overnight at 4 °C. Blots were washed in TBS-T (100 mM Tris-HCL pH 7.4, 150 mM NaCl, and 0.05% Tween-20) and incubated with secondary antibody in blocking buffer for 1 h at RT. Blots were washed in TBS-T and rinsed with DEMI-H_2_O before fluorescence scanning. Both fluorescence scanning and quantification were performed using the Odyssey CLx Western Blot Detection System (LI-COR biosciences, Lincoln, NE, USA). Each protein of interest was normalized against a reference protein on the same blot (GAPDH).

### 2.7. RNA Isolation

For RNA isolation, cerebral organoids were collected individually in 500 µL of TRIzol Reagent (Thermo Life Technologies, 15596018) and lysed using the Ultra Turrax Homogenizer (IKA, 0003737000). Samples were stored at −80 °C until further use. RNA was isolated using the miRNeasy mini kit (217004, Qiagen, Hilden, Germany) according to manufacturer’s instructions. Genomic DNA was removed by including a supplementary DNAse step (Qiagen, 79254). RNA concentrations were measured using the Qubit RNA HS assay kit (Q32852, Invitrogen, Waltham, MA, USA) on the Qubit 4 (Invitrogen, Q33238).

### 2.8. RNA Sequencing

#### 2.8.1. Bulk RNAseq Analysis

Preparation of the cDNA libraries was performed by Single Cell Discoveries (Utrecht, [40]) according to the CelSeq2 protocol [41]. As a pre-sequencing quality control, the quality of the amplified RNA and the final cDNA libraries were determined with RNA 6000 Pico Kit (5067-1513, Agilent Technologies, Inc., Santa Clara, CA, USA) or the RNA High Sensitivity Kit (Agilent, 5067-4626), respectively, using the Bioanalyzer 2100 (Agilent, G2939BA). Samples were processed and sequenced in 2 separate batches. Libraries were sequenced using the Illumina NextSeq 5000 platform with paired-end sequencing (75 bp) at a sequencing depth of 10 million reads per sample. After de-multiplexing of the sample libraries, raw RNAseq reads were aligned along the hg38 human RefSeq transcriptome via Burrows-Wheeler Aligner [42] using MapAndGo.

#### 2.8.2. Quality Control

Using R studio (version 4.0.0, R core team (2020), PBC, Boston, MA, USA) quality control of all sequenced samples together was performed based on the following metrics: ERCC spike-ins, library size, and mitochondrial RNA (mtRNA). ERCC spike-ins were low (<1%) in all samples, suggesting high endogenous purity of the RNA content. Samples with a library size below 5 × 10⁵ counts were excluded. Reads mapped to mitochondrial genes were included, as expression effects associated with metabolism (mTOR-related activity) were a potential relevant outcome measure. However, samples identified as distribution-based outliers were excluded. Altogether, this led to the exclusion of 6 samples in the long-term exposure group (3.50 mM AA-exposed organoids and 3 CTR organoids, of which 3 were from iPSC line OH2.6, 2 were from OH1.5, and 1 was from OH4.6). To reduce noise within the gene expression data, genes with expression lower than the 15th percentile of the count distribution were removed. This accounted for removal of approximately 20% of the genes [43].

#### 2.8.3. Pre-Processing and Confounders

Inter-sample distances were assessed for each time point separately (week 4, week 5, and week 15). We applied unsupervised hierarchical clustering with the pheatmap v1.0.2 package using Euclidean distances and principal component analysis (PCA). To dissect the sources of variance in our data, we calculated Pearson *r* correlation between covariates and principal components. The variance within each gene explained by each single covariate (*R*^2^) was obtained by fitting a variance partition model using variance partition v1.2.5 package [44] (Appendix A). These analyses suggested that batch and cell line accounted for a significant amount of the variance in our gene expression data. We therefore created a nested variable encompassing both batch and cell line, which we implemented in the model for our differential expression analysis, referred to as “experiment”. For plotting purposes, we transformed the raw counts with the variance stabilization function from the DESeq2 v.1.12.3 package [45]. To account for confounding effects during visualization post-normalization, we then took our final model (~log-transformed library size + percentage ERCC + experiment + exposure) and removed variance associated with these confounding variables while retaining variance stemming from exposure (unexposed vs. exposed) using the *removeBatchEffect* function from the limma v3.28.14 package. After correction for these covariates, there were no sample outliers, as assessed with PCA, intra-sample correlation heatmap, and interquartile range (Appendix A).

#### 2.8.4. Differential Gene Expression Analysis

Differential gene expression between control samples and AA-exposed samples (both 10 mM and 50 mM) was analyzed for each time point separately (week 4, week 5, and week 15) using the DESeq2 package function with the nested variable for batch and cell line as a covariate. Differentially expressed genes (DEGs) were visualized using volcano plots from the EnhancedVolcano package v1.7.16 [46] (Appendix A). Heatmaps and custom plots were generated with ggplot2 v3.3.2 [47].

#### 2.8.5. Ingenuity Pathway Analysis

Canonical pathway analyses were performed on the differentially expressed genes (Adj. *p* < 0.05) using the Ingenuity Pathway Analysis (IPA) software. The Ingenuity gene knowledgebase was used as the reference, using Fisher’s exact test with Benjamini–Hochberg (BH) correction to determine significant enrichment (*p* < 0.05). Significant pathways with at least 5 overlapping genes were selected. 

#### 2.8.6. Comparison with Previously Published Datasets

Differentially regulated genes were compared with previously published datasets on specific cell types in cerebral organoids [38] and human microglia [29] (Appendix A), and significant overlap between gene lists was determined using Fisher’s exact test with Benjamini–Hochberg correction.

#### 2.8.7. Data Availability

Relevant data supporting the discussed findings are included in the paper and its Appendix A. From the RNA sequencing analysis, the raw count matrices and the R-code are available through our GitHub repository (https://github.com/ar-kie/MKMD.git, accessed on 6 April 2022).

### 2.9. Weighted Gene Co-Expression Network Analysis

We generated signed co-expression networks from the gene expression data using the package CoExpNets v 0.1.0 [48]. Briefly, a topological overlap matrix (TOM) was created based on Pearson *r*-derived gene adjacencies with the WGCNA package v1.70.30 [49]. As input, we used a filtered and covariate-corrected gene matrix (see DGE analysis) containing all samples and genes. Iterative k-means were then applied to an unmodified TOM to generate centroid-based gene–gene clusters (instead of the conventional WGCNA hierarchical clustering based on Pearson *r*-derived “module membership”). Each cluster’s expression pattern was summarized by calculating its first principal component (module eigengene; ME). The MEs were then used to a. calculate each gene’s module membership (MM; ME–gene Pearson correlations), b. calculate module–module similarities (ME–ME Pearson correlations), and c. calculate module expression differences between AA exposure conditions (*t*-test of ME expression and ME–concentration Spearman correlations). MM values were leveraged to select the top hub genes of each module (highest *R*^2^). Hub genes have been shown to be biologically true drivers of their respective gene clusters [49]. All plots were generated in R using ggplot2 v3.3.5. Network graphs were generated using the GGally v2.1.2 and igraph v1.2.7 packages. 

### 2.10. Statistics

Statistical analysis and figure design were performed with Graphpad software (version 8.4) and R studio (version 4.0.0, R core team (2020), PBC, Boston, MA). Results are expressed as median and interquartile range (IQR) in boxplots or violin plots with independent data points. For Western blot experiments, unpaired Kruskal–Wallis test or Mann–Whitney U test, followed by Dunn’s tests for multiple comparisons, was used to compare differences between conditions. Two-way ANOVA with Dunnett’s multiple comparisons test was performed to test the differences in the size of the organoids over time. The effect of outliers was explored by identifying values +/− 1.5 the interquartile range from the upper or lower quantile. Fisher’s exact test was used to test the overlap between the different gene sets using Benjamini–Hochberg correction. The significance level was set to *p* < 0.05 (two-sided). 

## 3. Results

### 3.1. mTOR and P70S6K in Human Cerebral Organoids

Cerebral organoids were generated following a previously described protocol [33] (Figure 1). Baseline expression of mTOR and the downstream ribosomal P70-S6 kinase 1 (P70S6K) was present throughout organoid development (at week 4, week 5, and week 15) at the protein and mRNA levels (Figure 2A,B; Appendix A). The figures show both total and phosphorylated protein expression of mTOR and P70S6K to reflect phosphorylation states (phospho-mTOR/mTOR ratio and phospho-P70S6K/P70S6K ratio) as a proxy for mTOR activity. Over time, the phosphorylation state of mTOR remained stable (*H*(2) = 2.70; *p* = 0.27), but the phospho-P70S6K/P70S6K ratio was downregulated (*H*(2) = 9.26; *p* = 0.0097) at week 15 (mean rank week 4 = 16.50; mean rank week 15 = 6.25; *p* = 0.01). *mTOR* and *RPS6KB1* mRNA expression increased over time (*mTOR*: *H*(2) = 28,98; *p* < 0.001; mean rank week 4 = 10.24; mean rank week 5 = 35.00; mean rank week 15 = 22.77 − *RPS6KB1*: *H*(2) = 28.82; *p* < 0.0001; mean rank week 4 = 11.12; mean rank week 5 = 36.00; mean rank week 15 = 21.23). An acute effect of rapamycin (a potent inhibitor of mTOR) on mTOR signaling in cerebral organoids was identified in 4-week-old organoids. One hour of exposure to rapamycin strongly inhibited the phosphorylation state of P70S6K (CTR: median = 0.94, IQR = 0.87; rapamycin: median = 0.23, IQR = 0.17; *p* = 0.0022) (Figure 2C; Appendix A).

### 3.2. Acute Effects of AA Exposure on mTOR Activity in Human Cerebral Organoids

Since mTOR and its downstream target P70S6K are expressed in the cerebral organoid, and mTOR activity can be downregulated with rapamycin, we next investigated the acute effect of AA exposure on mTOR signaling in cerebral organoids. Four-week-old organoids were exposed to medium enriched with 10 mM or 50 mM of the amino acids His, Lys, and Thr for 1 h (Figure 2D,E). Acute AA exposure with 50 mM of the amino acids showed the potent downregulation of the phosphorylation of P70S6K (*H*(2) = 18.20; *p* = 0.0001; mean rank CTR = 16.33; mean rank 50 mM = 5.111; *p* = 0.0054) (Figure 2D,E; Appendix A). 

### 3.3. Chronic Amino Acid Exposure Causes Size Deficits in Cerebral Organoids

The cerebral organoid size was determined by measuring the 2D area. Cerebral organoids subjected to chronic AA exposure were significantly reduced in size, as shown by the significant interaction between the effects of time and AA exposure on the size of the organoids (F(8, 352) = 22.47; *p* < 0.0001) (Figure 3A). AA-exposed cerebral organoids were smaller after several days of exposure (end of week 4) in the 50 mM condition and from week 7 in the 10 mM condition (Dunnett’s multiple comparison; *p*-adj. < 0.01) (Figure 3A; Appendix A). Representative images (Figure 3B) and the quantification of all the individual data points (Figure 3C,D) at weeks 5 and 10 illustrate the size differences. 

### 3.4. Transcriptomic Differences after Amino Acid Exposure in Cerebral Organoids

To assess whether AA exposure induced changes in transcription, RNA sequencing was performed. Transcriptomic differences were studied between AA-exposed (10 mM and 50 mM exposure together) and CTR cerebral organoids (a total of *N* = 95) for acute (1 h at week 4 *N* = 40), prolonged (chronic exposure until week 5 *N* = 31), and long-term (chronic exposure until week 15 *N* = 24) exposure separately. The whole transcriptome analysis between AA-exposed and CTR cerebral organoids at the three different time points identified a total of 1659 DEGs (FDR-corrected *p*-value < 0.05), of which 1009 genes were upregulated, and 686 genes were downregulated (Figure 4A–C; Appendix A). DEGs were identified at each time point, with the largest number of DEGs represented at the prolonged (week 5) time point. Most of the DEGs were time-point-specific (Figure 4B).

### 3.5. Cell Type and Pathway Analyses of DEG

To determine potential cell-type-specific effects related to AA exposure, we compared our DEGs with two independent datasets on cerebral organoid cell types and core microglia [29,38]. Fisher’s exact test revealed significant enrichment of the DEGs of microglial genes (long-term; 17/249; *p* = 0.0018), astroglia genes (prolonged; 13/115; *p* = 0.0094 and long-term; 17/115; *p* < 0.0001), neuroepithelial genes (long-term; 11/113; *p* = 0.0005), and proliferative precursors (long-term; 17/74; *p* < 0.0001) (Figure 4D, Appendix A). To gain further insight into the biological processes affected by AA exposure, we performed pathway analysis using IPA on the significant DEGs for each time point. In line with the protein expression data, several regulated pathways were related to mTOR signaling (“mTOR signaling”, “Regulation of eIF4 and P70S6K signaling”, and “EIF2 signaling”). Among the other pathways are processes related to development (“Regulation of the epithelial mesenchymal transition in development pathway”) and immune-related pathways (“Autophagy”, “IL8 signaling”, “Complement System”, and “Chondroitin Sulfate Degradation”) (Figure 4E, Appendix A).

### 3.6. Weighted Gene Co-Expression Network Analysis

To further increase insight into the transcriptomic differences between AA-exposed and control cerebral organoids, we performed a weighted gene co-expression network analysis (WGCNA) on all time points together. This resulted in the identification of 27 modules with intercorrelating transcripts, each containing more than 380 genes (Figure 5A,B; Appendix A). We used the module eigengene to determine correlations between the AA exposure concentration and the module expression pattern (Figure 5C). We identified significant module eigengene associations with AA exposure for the tan and cyan modules (Figure 5C,D). Using IPA, we allocated biological pathways to these modules, “inositol signaling” and “EIF2 & mTOR signaling”, respectively, again underscoring the involvement of the mTOR pathway on a genetic level (Figure 5E, Appendix A). The top hub genes from both modules were mapped to illustrate the interaction networks (Figure 5F, Appendix A). The central hub gene in the “inositol signaling” (tan) module, HACD3, is involved in metabolism through enzymatic activity, creating long-chain fatty acids that function as precursors of membrane lipids and lipid mediators [50]. The “EIF2 & mTOR signaling” (cyan) module hub gene, RACK1, is involved in mTOR signaling and regulated by nutrient starvation-induced mTOR inhibition [51]. Next, overlaying the DEGs between AA-exposed and control organoids from the RNAseq analysis with the WGCNA modules revealed that several modules harbored the majority of these DEGs (Figure 4G, Table 1). The blue, cyan, light cyan, and white modules were enriched in downregulated DEGs, and the dark green, dark grey, dark orange, orange, tan, and yellow modules were enriched in upregulated DEGs (Table 1). Cell-type enrichment analyses and IPA were used to annotate these DEG modules (Figure 5G, Table 1, Appendix A). Genes with decreased expression in AA-exposed organoids were significantly enriched in modules related to cell growth (“EIF2 & mTOR signaling” and “DNA replication”) and the “Microglia” module. Furthermore, the “EIF2 & mTOR signaling” module was significantly enriched in proliferative progenitor genes. Genes with increased expression after AA exposure were represented in modules associated with inflammation (“Interleukin signaling”, “Neuroinflammation”, and “Astrocyte”), the “inositol signaling” module, and the “mitochondrial (dys)function” module. 

## 4. Discussion

This study reports on the effects of cerebral organoid exposure to increased levels of the amino acids His, Lys, and Thr on neurodevelopment. In cerebral organoids, AA exposure inhibited mTOR activity, caused the retention of size, and affected gene expression in developing cerebral organoids. This raises interest in regard to possible therapeutic applications.

In cerebral organoids, the mTOR signaling cascade was affected by exposure to rapamycin, similar to previous studies [25,26]. In response to AA exposure with 50 mM of His, Lys, and Thr, mTOR activity was potently inhibited (illustrated by a strong decrease in P70S6K phosphorylation), in agreement with in vitro models and rodent studies [25,26]. Exposure to lower concentrations of the specific amino acids (10 mM) was related to size changes but had no effect on P70S6K phosphorylation in the cerebral organoids, potentially due to the sensitivity of the assay (Western blot).

The finding of the substantially inhibited size of the cerebral organoids after AA exposure is consistent with the fact that mTOR is involved in cell proliferation, metabolism, and growth and underscores the impact of amino acids on the early stages of human brain development [18,20,52,53]. After 1 week of AA exposure, cerebral organoid size was impacted, and at 10 weeks, the cerebral organoid size was substantially smaller. Gene expression data are consistent with mTOR changes. This pathway, through P70S6K, plays a crucial role in protein translation at the ribosome, and its activity is associated with protein synthesis and cell proliferation [54,55]. Therefore, mTOR inhibition directly affects protein synthesis, which, in turn, can suppress cerebral organoid growth. 

An elaborate analysis of gene expression using RNA sequencing and WGCNA consistently pointed to the involvement of mTOR and the immune system. Evidence for the involvement of mTOR was present at several levels. Several identified pathways enriched in DEGs at both week 5 and week 15 are directly and indirectly associated with mTOR signaling. Furthermore, WGCNA identified two modules that were regulated by AA exposure and enriched in DEGs, the cyan and tan modules. The cyan module is enriched in genes involved in mTOR and EIF2 signaling. The tan module is defined as the inositol phosphate pathway, involved in processes such as membrane transport, cytoskeletal function, and plasma membrane signaling, which has also been associated with mTOR signaling and, even more interesting, with amino acid mTOR sensing [56,57]. Moreover, the cyan module is enriched in genes related to proliferative precursors. These RNA sequencing results, combined with the fact that the mTOR pathway is involved in proliferation [52,53] and the finding that cerebral organoids remain smaller after AA exposure, together point to an mTOR-regulated defect in proliferation during neurodevelopment in the AA-exposed cerebral organoids. In the same line, EIF2 signaling could also affect neuroepithelial proliferation, as it is involved in the coordination of cell metabolic status and adaptive response signaling, and its phosphorylation leads to the attenuation of protein synthesis. Interestingly, more and more signaling pathway interactions between mTOR and EIF2 are being identified [58,59,60,61].This perspective is consistent with the fact that outer radial glia (a class of precursor cells predominant in human cortical development) express high mTOR activity and are therefore expected to be extra vulnerable to decreases in mTOR activity, affecting their proliferative capacity [16,32]. It would be valuable to characterize this cell population in the cerebral organoid upon AA exposure with HOPX/PAX6 immunohistochemistry and their proliferation rate with a BrdU assay. 

Furthermore, the DEG and WGCNA results also point to the involvement of the immune system and identified enrichment for microglial and astrocytic genes after AA exposure; both glial cell types are involved in inflammation and are highly responsive to external stimuli [62,63]. Previous research showed that mTOR plays a role in the regulation of the gliogenic switch, and changes in mTOR activity during neurodevelopment can lead to a changed ratio between the number of neurons and astrocytes [17,19]. In addition, DEGs at week 15 were enriched in genes from the microglia-associated complement system, which has previously been associated with increased mTOR activity in tuberous sclerosis patients’ post-mortem brain material [64]. Finally, the results of the pathway analysis of the DEGs and WGCNA modules are consistent with the role of microglial and astrocytic immune function after AA exposure and identified processes related to inflammation: “Autophagy” [65], “IL8 signaling”, “Chondroitin Sulfate Degradation” [66], “Interleukin signaling”, and “Neuroinflammation”.

### 4.1. Future Perspectives

The fact that the results from this study are consistent with the finding from animal research that mTOR activity is decreased in the brain after dietary AA supplementation [26] opens new avenues of nutritional research into neurodevelopment. The cerebral organoid proves to be a valuable translational model between animal studies and humans and, because of its versatility, could be employed in many different paradigms of nutritional intervention.

This study contributes to the body of evidence relating dietary amino acid changes to mTOR inhibition and neurodevelopment. The results suggest that cerebral organoid models can be employed to identify human neurodevelopmental processes that are vulnerable to environmental impacts or can be targeted by (nutritional) interventions. Assessing cell-type-specific changes using immunohistochemistry and single-cell RNA sequencing can further advance this emerging field.

In the future, investigating the effect of amino acid-mediated mTOR inhibition on neurodevelopment can contribute to our understanding of dietary modifiers of neurodevelopment and, potentially, later-life cognition and the risk of acquiring neuropsychiatric conditions [1,2,3,4,5,6]. The results from this study suggest that prenatal dietary amino acid exposure to His, Lys, and Thr is detrimental to neurodevelopment; however, they also strengthen the idea that dietary amino acids may have potential as therapeutic applications for neurodevelopmental disorders related to mTOR function, such as autism spectrum disorder, schizophrenia, and tuberous sclerosis [23,67,68]. 

Altogether, the current study warrants further research in this direction and may provide leverage for new applications of in vitro cerebral organoid models in studies of neurodevelopmental disorders, as well as for pioneering the application of nutritional interventions.

### 4.2. Limitations

This study should be interpreted in the context of its limitations. In our study, we increased the concentrations of the amino acids His, Lys, and Thr to non-physiological levels (10 mM and 50 mM), while the metabolic capacity of the cerebral organoids is not entirely understood. However, the concentrations that we applied are in a similar millimolar range to those in previous in vitro studies targeting mTOR with these amino acids [25,26]. 

Another important limitation is the absence of an AA exposure group with different amino acids. Previously, it was shown that in vitro exposure to increased levels of the branched-chain amino acids leucine, isoleucine, and valine enhanced mTOR activation. It would be of interest to determine if a similar effect can be observed in cerebral organoids in future experiments.

Furthermore, the large size difference between AA-stimulated and control organoids may indicate a compromised state of the cerebral organoids after AA exposure. Although no toxicity analyses were performed, the gene expression data do not point to an increase in apoptosis or necrosis.

In addition, by including other downstream factors such as IRS-1, 4EBP1, AKT, and IP3K, we could have more elaborately dissected the specific pathway elements through which the amino acids influence mTOR signaling.

Cerebral organoid research in general comes with several challenges. Firstly, there is heterogeneity between cerebral organoids [38]. The cell-type composition of cerebral organoids is known to be variable. Due to extensive filtering of RNAseq data, cell types with low frequencies (such as microglia) will have a smaller impact on the analysis. Furthermore, we cannot rule out that differences in cell composition between conditions may have had an impact. Differences in cell-type composition between time points are expected, and by analyzing each time point separately, these differences will not have contributed to differentially expressed genes. In addition, studying three cell lines, from three separate donors, at three time points, and at three concentrations does involve a substantial number of organoids. Nevertheless, heterogeneity between organoids may require larger sample sizes to detect smaller differences. Secondly, although cerebral organoids resemble the developing human brain on transcriptomic, epigenetic, and structural levels [31,33,35,36], they do not mature in the same way or form the structural complexity found in the real human brain. Therefore, the model is primarily suited to study the effects of AA exposure on the early phases of human brain development. Lastly, it is important to consider that in vitro conditions of brain organoid models increase stress pathway activation, affecting cell subtype specification [69]. These differences in stress responses could have influenced our power to detect small differences between CTR and AA-exposed organoids.

## 5. Conclusions

In conclusion, our study shows for the first time that exposure to increased levels of the amino acids histidine, lysine, and threonine inhibits mTOR activity, decreases size, and affects gene expression pathways related to mTOR, proliferation, and immune function in a human model of neurodevelopment. The results echo findings in animal studies that His, Lys, and Thr inhibit mTOR activity in brain cells and support the potential of cerebral organoid models as additional model systems for nutritional studies of brain development. It confirms the impact of early nutritional availability of amino acids as a powerful moderator of mTOR function. Considering the role of mTOR in neurodevelopment and particularly in neurodevelopmental disorders such as autism, schizophrenia, and tuberous sclerosis, this study has the potential to facilitate further progress in the understanding of the etiology of these disorders and the development of early nutritional interventions.

## Figures and Tables

**Figure 1 nutrients-14-02175-f001:**
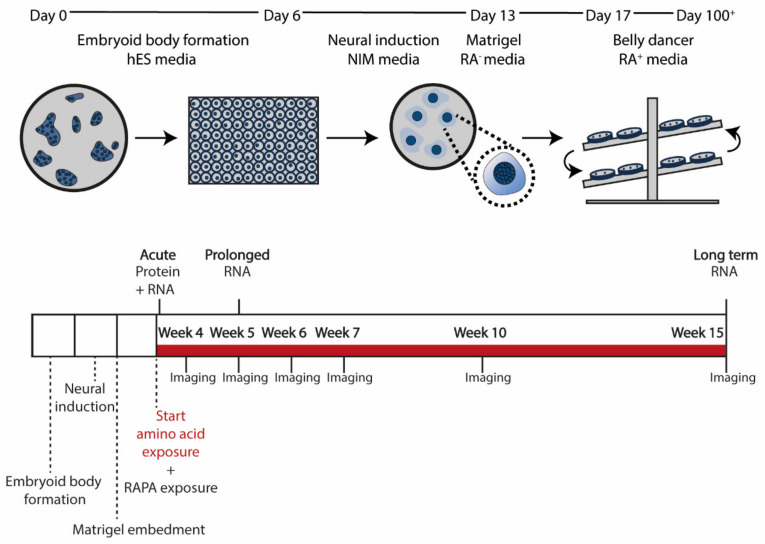
Methodological set-up. A schematic representation of the cerebral organoid protocol (based on Lancaster et al., 2013, with adaptations from Ormel et al., 2018) and experimental set-up. From the start of week 4, organoids were exposed to rapamycin, the AAs His, Lys, and Thr (10 mM or 50 mM), or control medium and either harvested after acute exposure (after 1 h) for protein and RNA isolation or kept in culture for chronic exposure. Organoid size was followed through bright-field imaging at the end of week 4, week 5, week 6, week 7, and week 10. Organoids were harvested for RNA isolation after prolonged (week 5) and long-term (week 15) AA exposure. Harvesting always occurred 1 h after medium change.

**Figure 2 nutrients-14-02175-f002:**
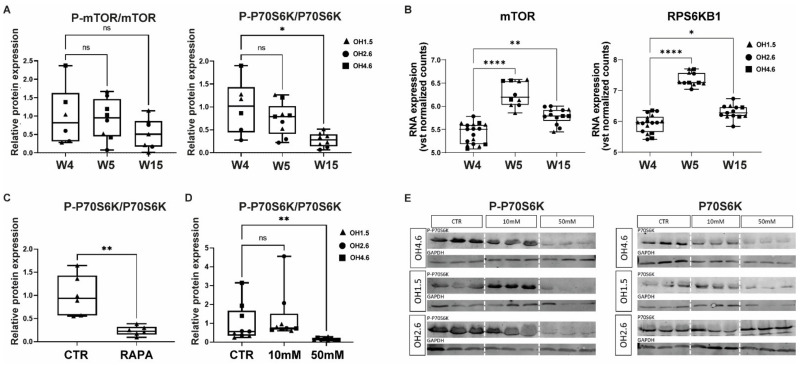
Baseline mTOR pathway expression and response to rapamycin and AA exposure in cerebral organoids. (**A**) Quantification of Western blots for the mTOR–phospho-mTOR (P-mTOR/mTOR) ratio and the P70S6K–phospho-P70S6K (P-P70S6K/P70S6K) ratio in CTR organoids at different time points (week 4, week 5, and week 15). (**B**) mTOR and P70S6K gene expression in CTR organoids showing differences in DESeq2 vst log-transformed raw counts at different time points (week 4, week 5, and week 15). (**C**,**D**) Quantification of Western blots for (**C**) P-P70S6K/P70S6k ratio in rapamycin-exposed organoids and (**D**) P-P70S6K/P70S6k ratio in acutely AA-exposed (10 mM/50 mM) organoids. (**E**) Western blots of acutely AA-exposed cerebral organoids lysates quantified in (**E**). Three samples per condition for 3 cell lines were used. Western blots were cropped to show the relevant bands (P70S6K, P-P70S6K, and GAPDH). Boxplots display median and IQR, with whiskers from minimum to maximum values. Data points represent individual organoids from different cell lines (circles = OH2.6; triangles = OH1.5; squares = OH4.6). Data were analyzed with the Kruskal–Wallis tests with Dunn test for multiple comparisons. Significant differences compared to W4 (for (**A**,**B**)) and to CTR (for (**D**)) are indicated with * *p* < 0.05; ** *p* < 0.01, **** *p* < 0.001. Mann–Whitney test was used to compare rapamycin treatment to CTR condition (** *p* < 0.01).

**Figure 3 nutrients-14-02175-f003:**
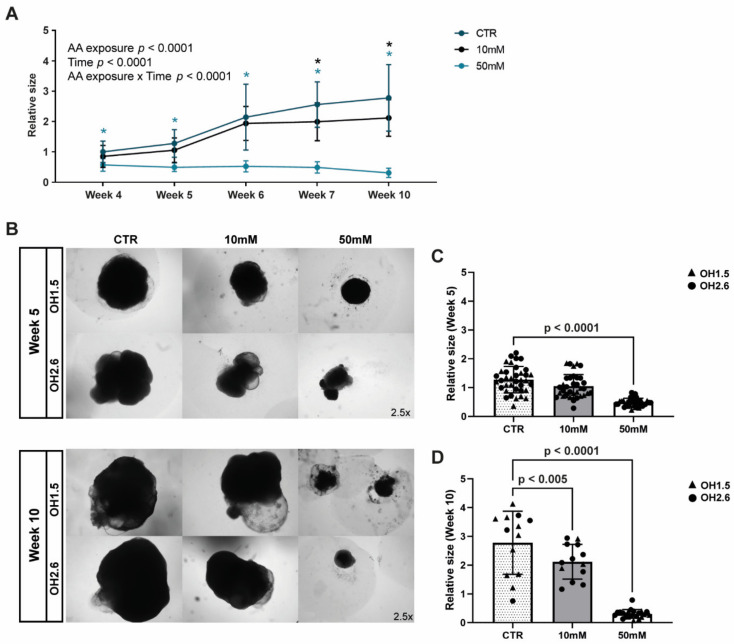
Growth responses to AA exposure in cerebral organoids. (**A**) Quantification of size measurements during 10 weeks of cerebral organoid development (mean ± SEM) for CTR, 10 mM AA-exposed, and 50 mM AA-exposed conditions. Data are expressed relative to the average of week 4 CTR cerebral organoid sizes within each cell line. *N* = 5–25 organoids per cell line (OH1.5 and OH2.6) per time point. Data were analyzed with a two-way ANOVA, from which *p* values are shown in the graph. Significant *p* values from post hoc Dunnett’s test for multiple comparisons are indicated by a blue asterisk for CTR versus 50 mM AA exposure (week 4–10 *p* < 0.0001) and a black asterisk for CTR versus 10 mM AA exposure (week 7 *p* = 0.0041; week 10 *p* = 0.0008), * *p* < 0.005. (**B**) Representative bright-field microscopy images (2.x magnification) of cerebral organoids at week 5 and week 10 of differentiation from 2 different cell lines (OH1.5 and OH2.6) for the different AA-exposed conditions (10 mM and 50 mM AA exposure and control). (**C**,**D**) Quantification of size measurements at week 5 (**C**) and week 10 (**D**) of cerebral organoid differentiation for CTR, 10 mM AA-exposed, and 50 mM AA-exposed conditions. Data are expressed relative to the average of week 4 CTR cerebral organoid sizes within each cell line. Bar graphs show mean ± SEM and individual data points (circles = OH2.6; triangles = OH1.5).

**Figure 4 nutrients-14-02175-f004:**
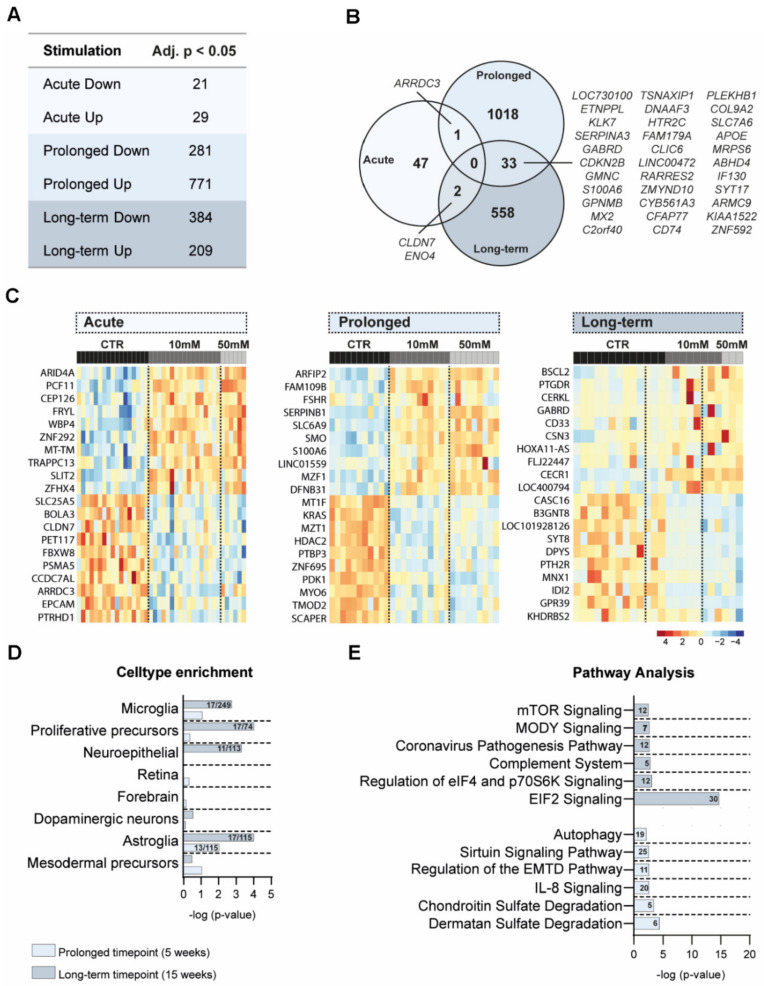
Altered gene transcription in response to AA exposure in cerebral organoids. (**A**) Table providing an overview of the number of significantly differentially expressed genes after acute, prolonged, and long-term AA exposure. (**B**) Venn diagram showing the overlap between significantly differentially expressed genes after acute, prolonged, and long-term AA exposure. (**C**) Heatmap representations of row-scaled DESeq2 vst transformed and adjusted z-scores of the most significantly (p-adj) differentially expressed genes (10 up/downregulated) for acute, prolonged, and long-term AA exposure. (**D**) Cell type analysis of differentially expressed genes after AA exposure at the 3 different time points, showing the overlap between cell-type-specific gene lists and the differentially regulated genes determined by Fisher’s exact test. At the acute time point, no significant cell-type enrichment was found. (**E**) Ingenuity pathway analysis (IPA) of differentially expressed genes after AA exposure of organoids at the different time points, showing only significant terms (the *x*-axis shows the significance (−log (*p*-value)). No significant enrichment of canonical pathways was found for differentially expressed genes at the acute time point.

**Figure 5 nutrients-14-02175-f005:**
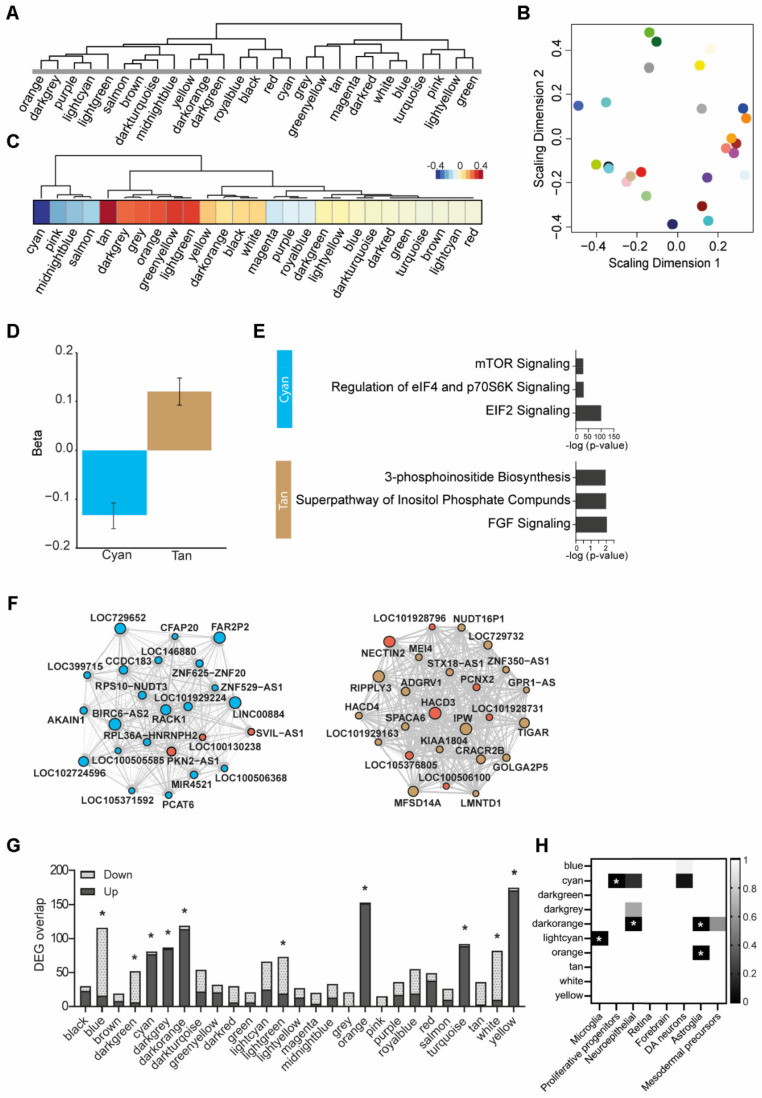
Gene network analysis of responses to AA exposure in cerebral organoids. (**A**) Tree cluster structure of WGCNA module generation. (**B**) Principal component analysis plot with ME distances, representing module similarity. (**C**) Module eigengene correlation with the factor “amino acid concentration”. (**D**) Significant module–eigengene associations with amino acid concentration are observed for the cyan (downregulated) and tan (upregulated) modules. (**E**) Top 3 enriched canonical pathways determined by Ingenuity pathway analysis (IPA) for the cyan and tan modules (the *x*-axis shows the significance (−log (*p*-value)). (**F**) Plots show top hub genes and their interaction network for the cyan and tan modules. (**G**) Overlap of down- and upregulated differentially expressed genes for each module as determined by Fisher’s exact test. Significant enrichment is indicated with *. (**H**) Heatmap showing cell-type enrichment of each module as determined by Fisher’s exact test. Significant enrichment is indicated with *, and scale bar indicates *q*-value.

**Table 1 nutrients-14-02175-t001:** Annotation of DEG-enriched WGCNA modules. The ten WGCNA modules that are significantly enriched in increased or decreased differentially expressed genes after AA exposure from RNA sequencing, using Fisher’s exact test with BH correction *p* < 0.05. Showing the annotation of each module based on cell type determination and IPA analysis, the enrichment of differentially expressed genes, module length, and the q-value from Fisher’s exact test.

Module	Annotation	Increased Expression		Decreased Expression	
		Enrichment	*q*-Value	Enrichment	*q*-Value
Blue	DNA replication			100/883	1.03 × 10^−21^
Cyan	EIF2 and mTOR signaling + proliferative precursors			46/382	6.44 × 10^−11^
Dark green	Interleukin signaling	54/689	1.62 × 10^−10^		
Dark grey	Neuroinflammation	77/620	5.04 × 10^−16^		
Dark orange	Undetermined	85/571	1.09 × 10^−18^		
Light cyan	Microglia			114/827	2.88 × 10^−6^
Orange	Astrocyte	152/801	3.5 × 10^−41^		
Tan	Inositol signaling	89/545	3.29 × 10^−19^		
White	Undetermined			72/890	1.04 × 10^−8^
Yellow	Mitochondrial (dys)function	171/853	7.2 × 10^−50^		

## Data Availability

Relevant data supporting the discussed findings are included in the paper and its Appendix A. From the RNA sequencing analysis, the raw count matrices and the R-code are available through our GitHub repository (https://github.com/ar-kie/MKMD.git, accessed on 6 April 2022).

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
