# Peer review of "Exposure to the Amino Acids Histidine, Lysine, and Threonine Reduces mTOR Activity and Affects Neurodevelopment in a Human Cerebral Organoid Model"

_nutrients, 2022, doi:10.3390/nu14102175_

Round 1

Reviewer 1 Report

The original paper "Nutrition shaping brain development: exposure to the amino acids histidine, lysine, and threonine reduces mTOR activity and affects neurodevelopment in a human cerebral organoid model" submitted by van Berlekom et al., showed the importance of nutrition in gestational life. Indeed, maternal diet during embryonic period it is crucial to brain development and to gut microbiota population. Although, future work it is need to clarify the role of this aa in early life, the present work help to understand the pathway to follow to next level of research.

Abstract: Well written and described the essence of manuscript

Introduction: Well written, explained very well the bibliography.

Methods: Very complete and detailed information

Results:

- Figure 1 - It is very small which makes it very difficult to read. I suggested that the authors divide figure 1 into two figures: one with only the experimental timeline, and the other with the results obtained by WB and PCR. Moreover, I suggested that authors put reference to Figure 1 in 2.4 Exposures sub-section. Legend it is ok.

- The first sentence of sub-section 3.3 Chronic amino acid exposure causes size deficits in cerebral organoids, it is very confuse. Please rewrite with small sentences.

- Figure 3 - panel C it is ok? or there are something wrong?

Discussion: 

Major points: 

-the authors did not shown toxicity studies following AA exposure. Please provide this information.

- It is also important provide information about proliferation. Did the authors do BrdU experiments?

Author Response

Comments and Suggestions for Authors

The original paper "Nutrition shaping brain development: exposure to the amino acids histidine, lysine, and threonine reduces mTOR activity and affects neurodevelopment in a human cerebral organoid model" submitted by van Berlekom et al., showed the importance of nutrition in gestational life. Indeed, maternal diet during embryonic period it is crucial to brain development and to gut microbiota population. Although, future work it is need to clarify the role of this aa in early life, the present work help to understand the pathway to follow to next level of research.
Abstract: Well written and described the essence of manuscript
Introduction: Well written, explained very well the bibliography.
Methods: Very complete and detailed information

We thank the reviewer for their positive assessment of the manuscript.

Results:

- Figure 1 - It is very small which makes it very difficult to read. I suggested that the authors divide figure 1 into two figures: one with only the experimental timeline, and the other with the results obtained by WB and PCR. Moreover, I suggested that authors put reference to Figure 1 in 2.4 Exposures sub-section. Legend it is ok.

According to the reviewers suggestion we have separated figure 1 into two figures and increased the Font sizes in the revised manuscript. We have also added a reference to Figure 1 in section 2.4.

- The first sentence of sub-section 3.3 Chronic amino acid exposure causes size deficits in cerebral organoids, it is very confuse. Please rewrite with small sentences.

We thank the reviewer for this comment. We have adjusted the sentence accordingly in the manuscript:

Cerebral organoid size was determined by measuring the measured by the 2D area. Cerebral organoids exposed to chronic AA exposure were significantly reduced in size as shown by the significant interaction between the effects of time and AA exposure on the size of the organoids (F(8, 352) = 22.47; p < 0.0001) (Figure 2A).”

- Figure 3 - panel C it is ok? or there are something wrong?

We have assessed panel C from figure 3 (the heatmaps showing the most significant differentially expressed genes for the acute prolonged and long term AA exposure). We did not identify any irregularities in this figure.

Discussion: 

Major points: 

-the authors did not shown toxicity studies following AA exposure. Please provide this information.

We agree with the reviewer that the absence of extensive toxicity analyses is a limitation of the study. However, as briefly pointed out in the discussion, our gene expression data did not indicate an increase in apoptosis or necrosis in our AA stimulated cerebral organoids. We have now emphasized this limitation and our argument more clearly in the limitations section of our manuscript (4.2):

“Furthermore, the large size difference between AA stimulated and control organoids can indicate a compromised state of the cerebral organoids after AA exposure. Although no toxicity analyses were performed, the gene expression data do not point to an increase in apoptosis or necrosis.”

- It is also important provide information about proliferation. Did the authors do BrdU experiments?

We thank the reviewer for their suggestion. We have incorporated this idea for future experiments in the discussion section as follows:

“It would be valuable to characterize this cell population in the cerebral organoid upon AA exposure by HOPX/PAX6 immunohistochemistry and their proliferation rate with a BrdU assay.”

Reviewer 2 Report

The manuscript “Nutrition shaping brain development: exposure to the amino acids histidine, lysine, and threonine reduces mTOR activity and affects neurodevelopment in a human cerebral organoid model” investigates the effects of a set of three essential amino acids on mTOR activity in vitro to study neurodevelopment activities using cerebral organoid.

The study is interesting as it used a novel approach to investigate brain development in vitro. It could add values to highlight the exogenous effects of specific amino acids on neuronal response as a mimic with organoids. 

The primary concern is that the objective and rationale of the study are poorly conceived as reads from the abstract and introduction. Authors must revise the justification and necessity of this study involving cerebral organoids treated with a specific set of amino acids. 

The primary need of the study seems to explore the mechanism of the facts that have been established by other pre-clinical and in vitro studies. Essentially, this study investigates the changes in gene expression, with enrichment in genes related to the response of synergistic inhibitory effects of threonine, histidine, and lysine when exposed to human cerebral organoids. 

The present study was undertaken based on the study that shows that inhibition of the mTOR pathway improves behavior and neuropathology in mouse models of ASD. The activated mTOR signalling in the brain is thought to contribute to the pathogenesis of ASD. The mTOR signalling primarily operates via two different pathways. While mTORC2 regulates the actin cytoskeleton through Akt and PKC, mTORC1 exclusively mediates cellular ‘‘nutrient-sensing.’’ Previous studies showed that amino acids such as thr, his, lys synergistically inhibited the synthesis of total proteins by suppression of mTORC1 activity and modulation of IRS-1 phosphorylation. 

Present work could not highlight and connect these issues well in their findings.

All nutrients under the health umbrella involve nutrition. The title “nutrition shaping….” is vouge. Must be specific for a research article. 

Line 456- it is inappropriate to state that the study mimics nutritional effects on human neurodevelopment. Instead, it could simulate the effects of the acute/chronic drug (AA exposure) on neurodevelopment, potential as a therapeutic application for neurodevelopmental disorders. The latter must be the primary purpose of the study. 

The brain is not an organ that supports any metabolic activities, including amino acids and protein received via nutrition. It is unlikely that the brain will be exposed selectively to a set of amino acids in vivo through nutritional mode. The author must revise and direct their narrative in the appropriate direction.

Line 40-49 author should bring maternal protein nutrition in this section instead of nutrition.

It reads from the narrative in several places (eg.line 512- 520) that the study intends to prove the point that organoid research can reduce the need for animal studies for nutritional intervention. The organoids are limited that do not involve system biology. Authors used twice to refer to their changes with rodent studies within the abstract also suggest the importance of system biology. On the other hand, stage-wise development involving changes in gene expression could be better studied with relatively closer organ development with organoids.  

The reality is that no nutritional invention for human needs is accepted without a randomized clinical trial involving a sizeable human subject. Therefore, neither organoid nor animal studies can advocate nutritional intervention for human health and disease. Thus, such narratives (eg.line 512- 520) undermine the study's power and divert this article in another direction.

Fig1 B Graph with P70S6 shows that the sample size (n) between Week 4 and week 15 is not comparative.

Multiple comparison data presentations are confusing here. **p<0.01 indicates a significance level not specific to a group. An ideal way to represent multiple groups can be done with like and unlike letters in the superscript. It is also applicable to another figure (for example, fig.2A).

Fig.2A, C, and D- all must have similar scales on Y-axis.

Fig.2 C and D significance symbols are missing;

Fig 3D-E Neuroepithelial proliferation could be the major response with these AA exposures via EIF2 signaling; 

Fig.4 is not apparent how it adds values to the core of the work; except E & H; 

Table 1 has a clear message that partly overlaps with this figure; 

The suppression of organoid growth is due to reduced protein synthesis, while neuroepithelial proliferation could be the major compensatory response in vitro with these AA exposures via EIF2 signalling. These should be discussed. 

Line 533, the limitation section has its limitation. It needs to be revised. Although the study used adequate quality control in gene expression analysis, it has a major limitation: no positive or negative controls in the main experimental groups. Why in vitro study did not include other amino acids as a negative control? Organoid models mimic the real brain but don’t mature as normal brain cells do. Unlike the brain, structural complexity is not reflected in the organoids. Moreover, organoid cells have inappropriate activation of several cellular stress programs. All these should be placed as a limitation of organoid research.

Author Response

Comments and Suggestions for Authors

The manuscript Nutrition shaping brain development: exposure to the amino acids histidine, lysine, and threonine reduces mTOR activity and affects neurodevelopment in a human cerebral organoid model” investigates the effects of a set of three essential amino acids on mTOR activity in vitro to study neurodevelopment activities using cerebral organoid. The study is interesting as it used a novel approach to investigate brain development in vitro. It could add values to highlight the exogenous effects of specific amino acids on neuronal response as a mimic with organoids.  

We thank the reviewer for this positive assessment of our manuscript.

The primary concern is that the objective and rationale of the study are poorly conceived as reads from the abstract and introduction. Authors must revise the justification and necessity of this study involving cerebral organoids treated with a specific set of amino acids. The primary need of the study seems to explore the mechanism of the facts that have been established by other pre-clinical and in vitro studies. Essentially, this study investigates the changes in gene expression, with enrichment in genes related to the response of synergistic inhibitory effects of threonine, histidine, and lysine when exposed to human cerebral organoids. 

We thank the reviewer for raising this important aspect of the study and write-up . We agree that a balance is to be found here. While the reviewer is completely correct in the description of the essence of the work: “investigates the changes in gene expression, with enrichment in genes related to the response of synergistic inhibitory effects of threonine, histidine, and lysine when exposed to human cerebral organoids” the question whether this work is informative of cerebral responses to a dietary intervention remains warranted. In  order to avoid overstatements we have revised the manuscript at several points (including removing “nutrition” from the title).

In the introduction:

“This study pioneers in researching the effect of His, Lys, and Thr exposure on mTOR activity in a human neurodevelopmental 3D in vitro model. The primary need for the study is two sided: 1. Establish if we can replicate finding on mTOR activity upon His, Lys, and Thr exposure from previous in vitro and in vivo work using the human cerebral organoid model. 2. Explore neurodevelopmental responses to the inhibitory effects of His, Lys, and Thr on mTOR activity in human cerebral organoids by investigating changes in general size and gene expression. The developing cerebral organoids are exposed to increased concentrations of the 3 amino acids His, Lys, and Thr (AA exposure) and assessed for mTOR activity by western blot, general organoid size, and transcriptomic alterations using RNA-sequencing. The data are cross-referenced with gene-lists on biological processes and cell types to determine leads for future research.”

We have also adjusted part of the abstract for clarity:

“Evidence of the impact of nutrition on human brain development is compelling. Previous in vitro and in vivo work shows that three specific amino acids, histidine, lysine, and threonine synergistically inhibit mTOR activity and behaviour. Therefore, prenatal availability of these amino acids could be important for human neurodevelopment. However, methods to study the underlying mechanisms in a human model for neurodevelopment are limited. Here we pioneer the use of human cerebral organoids to investigate the impact of amino acid supplementation on neurodevelopment”

The present study was undertaken based on the study that shows that inhibition of the mTOR pathway improves behavior and neuropathology in mouse models of ASD. The activated mTOR signalling in the brain is thought to contribute to the pathogenesis of ASD. The mTOR signalling primarily operates via two different pathways. While mTORC2 regulates the actin cytoskeleton through Akt and PKC, mTORC1 exclusively mediates cellular ‘‘nutrient-sensing.’’ Previous studies showed that amino acids such as thr, his, lys synergistically inhibited the synthesis of total proteins by suppression of mTORC1 activity and modulation of IRS-1 phosphorylation. Present work could not highlight and connect these issues well in their findings.

We thank the reviewer for highlighting this and as the description is very useful took the liberty of paraphrasing.

We have revised the introduction:

"Amino acids are a key component of nutrition and some essential amino acids need to be provided through the diet. Dietary intake influences plasma amino acid concentrations and ratios[11] and supplementation of specific amino acids is used to improve health, metabolism, and athletic performance[12]. Amino acids are best known as the building blocks of proteins, but also have an important regulatory function in the cell[12].
Recently, amino acids have emerged as potent modulators of the mammalian target of rapamycin (mTORC)[15–17]. mTOR signaling regulates the phosphorylation of the translational modulator P70S6K and is involved in processes such as cell growth, metabolism, and autophagy, but also in neurodevelopment, regulating cortical structure formation through outer radial glia, timing of the gliogenic switch, and axon formation and dendritic arborization[18–22]. Deregulation of mTOR function, due to genetic mutations or altered protein expression is involved in brain diseases, particularly developmental neuropsychiatric disorders such as autism, schizophrenia, and tuberous sclerosis[18,23–26]. mTOR signaling operates via two pathways; mTORC1 which mediates cellular ‘‘nutrient-sensing’’ and mTORC2 which regulates the actin cytoskeleton through Akt and PKC.
Previous studies showed that supplementation of 3 specific amino acids, Histidine (His), Lysine (Lys), and Threonine (Thr), synergistically potently inhibit P70S6K signalling downstream of mTORC1 and thereby protein synthesis, and modulate IRS-1 phosphorylation[13,14 add reference to Prizant and Barash 2008 J. Cell Biochem.]. Furthermore, a mouse study enriching the postnatal diet of 5 week old mice with these 3 essential amino acids found decreased mTOR activity in the mice brains and an effect on autism related behaviours[13].
Therefore, amino acid availability and especially of His, Lys, and Thr at early developmental stages, is important for healthy brain development and dietary changes during pregnancy may have consequences for risk of neurodevelopmental disorders.”

Unfortunately this study only measured P70S6K as a measurement of mTOR activation. As a consequence, it is not possible to more elaborately dissect the specific pathway elements trough which mTOR amino acid regulation functions as would have been when we would have included  factors such as IRS-1 or 4EBP1, AKT, IP3K, etc. We now point this out in the limitation section of the discussion (4.2):

“Also, by including other downstream factor such as IRS-1, 4EBP1, AKT, IP3K, we could have more elaborately dissected the specific pathway elements through which the amino acids influence mTOR signaling.”

All nutrients under the health umbrella involve nutrition. The title “nutrition shaping….” is vouge. Must be specific for a research article. 

After consideration (see also point 1) we have rephrased the title as:

“Exposure to the amino acids histidine, lysine, and threonine reduces mTOR activity and affects neurodevelopment in a human cerebral organoid model”  

Line 456- it is inappropriate to state that the study mimics nutritional effects on human neurodevelopment. Instead, it could simulate the effects of the acute/chronic drug (AA exposure) on neurodevelopment, potential as a therapeutic application for neurodevelopmental disorders. The latter must be the primary purpose of the study. 

In accordance with the previous responses we  have rephrased the beginning of the discussion to tune down the nutritional aspect and introduce the potential therapeutic aspects:

“This study reports on the effects of cerebral organoid exposure to increased levels of the amino acids His, Lys, and Thr on neurodevelopment. In cerebral organoids AA exposure inhibited mTOR activity, caused a retention in size, and affected gene expression in the developing cerebral organoids. This raises interest in regard to possible therapeutic applications.”

The brain is not an organ that supports any metabolic activities, including amino acids and protein received via nutrition. It is unlikely that the brain will be exposed selectively to a set of amino acids in vivo through nutritional mode. The author must revise and direct their narrative in the appropriate direction.

This is an important point. Previous research has shown that dietary intake (for example the ratio of meat/fish/vegetables) affects amino acid plasma levels in humans as described in the introduction of our manuscript (line 51). We agree with the reviewer that in vivo the brain will never be exposed to a specific set of amino acids, however, that is not the paradigm applied in our study, as we enriched the three amino acids by increasing their concentration. Increasing plasma concentrations of specific amino acids in vivo trough diet or supplementation is realistic. The previous rodent study to which we refer several times in our manuscript (Wu et al., 2017), were able to affect mTOR activation in the brain by changing amino acid ratio’s in the food of the rodents.

To clarify we have adjusted the limitations section of our manuscript:

In our study we increased concentrations of the amino acids Thr, His, and Lys to non-physiological levels (10 mM and 50 mM) while the metabolic capacity of the cerebral organoids is not entirely understood”

Line 40-49 author should bring maternal protein nutrition in this section instead of nutrition.

We thank the reviewer for this observation and have adjusted the manuscript accordingly.

“Compelling evidence from epidemiological studies suggests that maternal diet during pregnancy is a key modifier of neurodevelopment and impacts later life intelligence, social function, and the risk for acquiring a range of neuropsychiatric conditions such as autism spectrum disorders and schizophrenia[1–6]. Particularly during early gestation, brain development is vulnerable to maternal nutritional deviations with effects persisting later in life. Pre-clinical studies on prenatal nutrition, availability of micronutrients, and composition of the maternal diet, have shown a broad range of effects in offspring, such as decreased neurogenesis, changes in neuronal dendritic arborization, and increased astrocytic GFAP expression[7–10].”

It reads from the narrative in several places (eg.line 512- 520) that the study intends to prove the point that organoid research can reduce the need for animal studies for nutritional intervention. The organoids are limited that do not involve system biology. Authors used twice to refer to their changes with rodent studies within the abstract also suggest the importance of system biology. On the other hand, stage-wise development involving changes in gene expression could be better studied with relatively closer organ development with organoids. The reality is that no nutritional invention for human needs is accepted without a randomized clinical trial involving a sizeable human subject. Therefore, neither organoid nor animal studies can advocate nutritional intervention for human health and disease. Thus, such narratives (eg.line 512- 520) undermine the study's power and divert this article in another direction.

We thank the reviewer for this point of critique. We think it is very important that our data resemble the findings of in vivo work, and this strengthens the legitimacy of using cerebral organoids for these types of studies, with the benefit of being human cells and therefore interesting from a translational perspective. We have now rewrote some parts in the abstract, discussion, and conclusion section to resonate more with this argumentation instead of focusing on “replacing animal studies”.

“The fact that the results from this study are consistent with the finding from animal research that  mTOR activity is decreased in the brain after dietary AA supplementation[13] opens new avenues of nutritional research into neurodevelopment. The cerebral organoid proves a valuable translational model between animal studies and humans, and because of its versatility could be employed in many different paradigms of nutritional intervention.”

Fig1 B Graph with P70S6 shows that the sample size (n) between Week 4 and week 15 is not comparative.

The sample sizes indeed differ for week 4 we had 6 samples (2 of each line) and for week 15 we had 8 samples (6 samples of OH1.5 and 2 samples of OH2.6). This does not interfere with the legitimacy of the analysis.

Multiple comparison data presentations are confusing here. **p<0.01 indicates a significance level not specific to a group. An ideal way to represent multiple groups can be done with like and unlike letters in the superscript. It is also applicable to another figure (for example, fig.2A).

Although we agree with the reviewer there are other ways to represent multiple comparison data, we propose to represent our data in a similar way as found, for example, in Blair et al., 2018 Nature Medicine (DOI: 10.1038/s41591-018-0139-y) (for example Figure 1D and 2H). All the statistical information for these figures is also provided in supplementary table 1 and 2)

We have now improved the description of our figures, clarifying the data presentations:

(Previous Figure 1 now) Figure 2 legend:

“Data were analysed with the Kruskall-Wallis tests with Dunn test for multiple comparisons. Significant differences compared to W4 (for Figure A and B) and to CTR (for Figure D) are indicated with * p < 0.05; ** p < 0.01. Mann-Whiteny test was used to compare rapamycin treatment to CTR condition (** p < 0.01).”

(Previous Figure 2 now) Figure 3 legend:

“Data were analysed with a two-way ANOVA, from which p-values are shown in the graph. Significant p values from post-hoc Dunnett’s test for multiple comparisons are indicated by a blue asterisk for CTR versus 50mM AA exposure (week 4-10 p < 0.0001), and a black asterisk for CTR versus 10mM AA exposure (week 7 p = 0.0041; week 10 p = 0.0008).”

Fig.2A, C, and D- all must have similar scales on Y-axis.

We have adjusted the figure (now figure 3) according to the reviewers suggestion.

Fig.2 C and D significance symbols are missing;

According to the reviewers suggestion we have added the p values to (now) figure 3 C and D.

Fig 3D-E Neuroepithelial proliferation could be the major response with these AA exposures via EIF2 signaling; 

We thank the reviewer for pointing this out to us. We have addressed this in the answer 2 points below.

Fig.4 is not apparent how it adds values to the core of the work; except E & H; Table 1 has a clear message that partly overlaps with this figure; 

As we understand, the author confirms the WGCNA adds value for the manuscript and the comment is directed primarily at the representation of these data. After careful consideration we propose to not alter (now) figure 5. Part A-D contain important background information which is necessary for transparency of our approach particularly for those with interest in WGCNA. We propose to retain  figure F because it these genes are the most important factors of the two identified modules, and readers might recognize genes or patterns that are not readily apparent or known to us. Showing this graphically has a preference over an additional table as it also gives an indication of the interactions between the genes. We agree with the reviewer that par G contains overlapping information with table 1, however figure 5G also shows the non-significant modules which gives some additional context.

The suppression of organoid growth is due to reduced protein synthesis, while neuroepithelial proliferation could be the major compensatory response in vitro with these AA exposures via EIF2 signalling. These should be discussed. 

This is again a good point. We have now emphasized the role of mTOR in protein synthesis as an explanation for reduced organoid size in the discussion:

“The find of substantially inhibited size of the cerebral organoids after AA exposure is consistent with the fact that mTOR is involved in cell proliferation, metabolism, and growth and underscores the impact of amino acids on the early stages of human brain development[18,19,53,54]. After 1 week of AA exposure cerebral organoid size was impacted, and at 10 weeks the cerebral organoid size was substantially smaller. Gene expression data was consistent with mTOR changes. This pathway, trough P70S6K, plays a crucial role in protein translation at the ribosome, and its activity is associated with protein synthesis and cell proliferation[55,56]. Therefore, mTOR inhibiton directly affects protein synthesis, which in turn is likely to suppress cerebral organoid growth.”

We have added a discussion on EIF2 signaling involvement in the discussion section:

“In the same line, EIF2 signaling could also affect neuroepithelial proliferation as it is involved in coordination of cell metabolic status, adaptive response signaling, and its phosphorylation leads to attenuation of protein synthesis. Interestingly, more and more signaling pathway interactions between mTOR and EIF2 are identified [add reference to Wengrod et al., 2015 Sci Signal; Misra et al., 2021 Nucleic Acids Res.; Klann et al., 2020 Mol. Cel.; Parveen et al., 2021 BioRxiv].”

Line 533, the limitation section has its limitation. It needs to be revised. Although the study used adequate quality control in gene expression analysis, it has a major limitation: no positive or negative controls in the main experimental groups. Why in vitro study did not include other amino acids as a negative control? Organoid models mimic the real brain but don’t mature as normal brain cells do. Unlike the brain, structural complexity is not reflected in the organoids. Moreover, organoid cells have inappropriate activation of several cellular stress programs. All these should be placed as a limitation of organoid research.

We thank the reviewer for this critical evaluation, and must agree. Therefore we have now incorporated the points mentioned by the reviewer in the limitations section of our discussion (4.2).

“Another important limitation is the absence of a AA exposure group with different amino acids. Previously it was shown that in vitro exposure to increased levels of the broad chain amino acids Leucine, Isoleucine, and Valine enhanced mTOR activation. It would be of interest to determine if a similar effect can be observed in the cerebral organoids in future experiments.”

“Secondly, although cerebral organoids resemble the developing human brain on transcriptomic, epigenetic, and structural level[30,34–36], they do not mature the same or form the structural complexity as found in the real human brain. Therefore, the model is primarily suited to study the effects of AA exposure on the early phases of human brain development. Lastly, it is important to consider that in vitro conditions of brain organoid models increase the stress pathway activation, affecting cell subtype specification [insert reference Bhaduri et al. 2020 Nature]. These differences in stress responses could have influenced our power to detect small differences between CTR and AA exposed organoids.”

Round 2

Reviewer 1 Report

The author answered my questions and improved the manuscript following my comments.

Reviewer 2 Report

Despite its limitations without negative control in the experimental design, the study certainly adds value to the existing knowledge.

The authors have made a reasonable effort in revising their work.